# Study Protocol for Genomic Epidemiology Investigation of Intensive Care Unit Patient Colonization by Antimicrobial-Resistant ESKAPE Pathogens

**DOI:** 10.3390/mps8060151

**Published:** 2025-12-13

**Authors:** Andrey Shelenkov, Oksana Ni, Irina Morozova, Anna Slavokhotova, Sergey Bruskin, Denis Protsenko, Yulia Mikhaylova, Vasiliy Akimkin

**Affiliations:** 1Central Research Institute of Epidemiology, Novogireevskaya Str., 3a, 111123 Moscow, Russiavgakimkin@yandex.ru (V.A.); 2Moscow Multidisciplinary Clinical Center «Kommunarka», Sosenskiy Stan Str., 8c3, 108814 Moscow, Russia; 3Institute for Globally Distributed Open Research and Education (IGDORE), 109651 Moscow, Russia; 4Vavilov Institute of General Genetics, Russian Academy of Sciences, Gubkina Str., 3, 119991 Moscow, Russia

**Keywords:** antimicrobial resistance, intensive care units, ESKAPE pathogens, bacterial colonization, whole genome sequencing

## Abstract

ESKAPE bacteria are a major global threat due to their rapid antibiotic resistance acquisition and severe healthcare-associated infections. Effective countermeasures require epidemiological surveillance and resistance transmission studies, particularly for antimicrobial-resistant (AMR) colonization in intensive care unit (ICU) patients. Whole-genome sequencing (WGS) provides critical information on resistance spread and mechanisms. In the provided protocol, rectal and oropharyngeal swabs, or endotracheal aspirate/bronchoalveolar lavage for intubated patients, are collected at ICU admission and twice weekly. Patient interviews and medical records identify risk factors for resistant microflora. Samples undergo cultivation, species identification, antibiotic susceptibility testing, and DNA extraction. Sequencing is performed using second- and third-generation platforms, with selected isolates subject to hybrid genome assembly. Resistance genes, virulence factors, and typing profiles (MLST, cgMLST) are determined. This protocol characterizes the ICU patient colonization by AMR pathogens, including species distribution, phenotypic and genotypic resistance profiles, clonal structure, and temporal changes. It estimates detection frequency and colonization patterns at each locus, identifies key risk factors, including prior community or inter-facility exposure, and analyzes associations between risk factors and admission colonization. The study aims to estimate AMR infection risk and severity in ICU patients through the comprehensive analysis of colonization dynamics, resistance patterns, and clonal characteristics using WGS data on pathogen composition and AMR trends.

## 1. Introduction

In the last decade, the antibiotic resistance of pathogenic and opportunistic bacteria has grown dramatically in all regions of the world [1,2,3,4]. The bacterial strains resistant to antimicrobials usually involved in last-line therapy, like carbapenems and polymyxins, are especially dangerous in clinical settings for immunocompromised patients and were shown to increase morbidity and mortality among patients [5,6]. Among these strains, the species belonging to the ESKAPE group, including *Enterococcus faecium*, *Staphylococcus aureus*, *Klebsiella pneumoniae*, *Acinetobacter baumannii*, *Pseudomonas aeruginosa*, and *Enterobacter* spp., have been shown to acquire antibiotic resistance with increasing rates, and searching for effective treatment strategies for these pathogens was assigned a critical priority by the World Health Organization [7].

Potentially successful approaches to beat back this global challenge could include proper antibiotic stewardship [8], developing novel antibiotics and alternative drugs or compounds [9,10,11], accelerating diagnostic procedures [12,13], and, last but not the least, taking appropriate prevention and epidemiological measures against rapid pathogen spreading within a particular healthcare facility, region, or country [14,15]. The achievement of the last three goals can be facilitated by the recently advanced area of research called genomic epidemiology. This field includes the practice of utilizing whole-genome sequencing (WGS) to access, index, and analyze DNA sequence features of epidemiologic importance [16]. Genomic epidemiology has already been applied for outbreak investigations [17,18], revealing the predominant antibiotic resistance determinants [19,20] and elucidating the possible ways of resistance transmission [21,22,23] for various bacterial, as well as viral [24], pathogens.

However, since the rates of bacterial genome change and horizontal gene transfer are rather high in the ESKAPE group [25,26], and many resistance and virulence gene transfer mechanisms still remain unclear [27], continuous genomic epidemiology surveillance is required on all levels of public healthcare. In addition, the exponential growth of available genomic data poses a lot of new questions to answer, for example, how to define bacterial strains in a post-genomic era [28,29] and more practical issues—e.g., how to develop a reliable procedure for distinguishing strains/isolates that have caused a particular outbreak. Although some strain definitions using WGS data have been proposed [30], currently they are not generally accepted.

At the same time, it is vitally important to investigate patient colonization by antimicrobial-resistant (AMR) bacteria, especially for patients with severe conditions who require treatment in special facilities like intensive care units (ICU). Particularly important is the detection of multidrug-resistant (MDR) bacterial isolates, which include the ones being resistant to three or more antibiotic classes. In order to develop advanced infection prevention strategies in ICUs, it is essential to continuously monitor the bacterial colonization of several loci starting from the moment of ICU admission and during patient treatment in this facility. Numerous reports have shown the dramatic increase in the rates of nosocomial infections in general, and the ones caused by AMR pathogens in various ICUs located in all parts of the world [31,32,33,34]. Thus, it is crucial to estimate the risk of AMR infection development and to rank its possible severity for all patients admitted to ICUs. However, to the best of our knowledge, appropriate, straightforward, and reliable study protocols concerning AMR monitoring in ICUs using WGS have not been established.

The proposed study protocol will facilitate addressing the issues stated above and developing better prevention and control strategies for healthcare facilities.

## 2. Materials and Methods

### 2.1. Study Design

This is an observational, experimental medicine study using samples taken from human participants recruited from a hospital ICU. The study includes four stages, which are presented in Figure 1. Briefly, the study will include the initial examination of antimicrobial resistance carriage within the microflora of patients admitted to the ICU, continuous monitoring of the same within the period of ICU stay, final examination upon patient discharge from the ICU, and data analysis.

### 2.2. Ethics Approval

Approval has been received from the Ethics Committee of Moscow Multidisciplinary Clinical Center “Kommunarka” (Moscow, Russia) on 12 April 2022 (protocol no. 4).

Informed consent will be obtained in written form from all participants upon hospital admission. These consents will include the permission to use the participants’ information for research, reporting, and data sharing purposes after removing personal information.

The results of the data analysis will be published in peer-reviewed international journals, and the genomic data for all bacterial isolates obtained through WGS will be deposited to public databases (Genbank and/or BIGSdb-Pasteur).

### 2.3. Study Population and Duration

The study will include at least 250 patients upon admission and during their stay in the ICU. Inclusion and exclusion criteria are given below in Table 1. This lower boundary was selected based on the previous epidemiological studies [35] in order to capture the variation in risk factors.

The initial duration of the study is intended to be 3 months, which will allow recruiting the required number of participants according to the previous ICU occupation and stay duration data [35]. The extension up to 6 months in total can be made, or the study can be terminated earlier when a sufficient amount of data is obtained. Bacterial culturing, DNA extraction, and sequencing will be performed during the study, with a possible extension of up to one month upon sample collection completion. Currently, no data are collected and no patient recruitment has been performed.

The start of the tentative recruitment period for this study is scheduled for 22 March 2026, while the end of the period is intended to be 22 June 2026.

### 2.4. Samples

Biological samples from the patients will be collected using transport swabs with Amies charcoal Medium (DELTALAB, Rubi, Spain). Particular sample localizations will include rectal and oropharyngeal swabs or endotracheal aspirate/bronchoalveolar lavage for the intubated patients. No additional samples will be taken and no interventions will be made except for the ones routinely obtained for prescribing correct antibiotic treatment.

### 2.5. Sample and Metadata Collection

#### 2.5.1. Upon Admission

In order to determine the AMR microflora carriage, biological samples listed above will be taken from every patient upon admission to the ICU. In addition, all the patients will be questioned and/or available medical records will be analyzed to find out the risk factors for the isolation of resistant microflora (Table 2). In addition, the condition severity for all patients will be assessed upon admission using the scores and criteria listed in Table 2. Finally, the presence of certain risk factors and comorbidities listed in Table 2 will be registered for all patients upon admission.

Missing data will be handled according to the mechanism of missingness and proportion of missing values per variable. For variables with <5% missing data (antibiotic exposure, invasive procedures), complete-case analysis will be performed. For variables with 5–20% missing data, multiple imputation by chained equations (MICE) will be applied using auxiliary variables, including pharmacy records, resistance patterns, and severity indicators; 30 imputed datasets will be generated and pooled estimates reported with standard errors accounting for imputation uncertainty. Variables with >20% missing data will undergo sensitivity analyses comparing optimistic (missing = low-risk assumption) and pessimistic (missing = high-risk assumption) scenarios to assess robustness of conclusions.

Secondary predictors (severity scores, comorbidities) with <10% missing data will be imputed using regression of related clinical parameters. For duration variables (hospitalization days, ICU stay, mechanical ventilation), structural zeros (inherently missing for non-intubated patients) will be distinguished from true missing data and handled by creating categorical indicators rather than imputation. Missing comorbidity data will be assumed absent (conservative bias) if not documented in discharge summaries, with verification through cross-reference of laboratory results and medication lists.

A missing data flow diagram documenting the missingness pattern by variable and ICU site will be generated. Complete-case analysis will be performed in parallel with imputation-based analyses to verify consistency and assess sensitivity to missing data assumptions. All results will be reported with explicit documentation of the imputation methodology and uncertainty estimates.

#### 2.5.2. During ICU Stay

The biomaterial will be taken from all patients on Mondays and Thursdays for colonization monitoring purposes. The loci from which the samples will be taken are the same as the ones upon admission (rectal, oropharyngeal swabs or endotracheal aspirate/bronchoalveolar lavage for intubated patients). The severity of the condition will be assessed according to the characteristics listed in column 2, Table 2 for all the patients at each biomaterial sampling. In addition, features of the course, including the necessity for special medicine and treatment, will be noted. When MDR microflora from a certain locus are detected, further monitoring of the microflora isolated from this locus will be performed only once a week on Mondays. At the same time, monitoring of the microflora from other loci in this patient will continue twice a week.

#### 2.5.3. Upon Completion of Hospitalization/Discharge from ICU

At the end of the patient’s treatment in the ICU, monitoring will be completed regardless of the previous isolation of the AMR microflora. For each patient, the severity of the course and the outcome of the disease will be assessed, including the length of the ICU stay, duration of hospitalization, the development of nosocomial infection, and the period of development. The antibacterial therapy prescribed to the patient will also be recorded.

### 2.6. Sample Processing

#### 2.6.1. Bacteria Cultivation, Identification, and Susceptibility Testing

Bacterial cultivation, sample preparation, and other processes will be performed according to the Russian recommendations on the determination of antimicrobial drug susceptibility for microorganisms, version 2025-01 [36], which are based on the corresponding recommendations by the European Committee on Antimicrobial Susceptibility Testing (EUCAST) [37].

The collected samples from each locus will be cultivated through a semi-quantitative sectoral method using the following solid media: Columbia blood agar base EP/USP/ISO and Urinary Tract Infection Chromogenic Agar (UTIC). Bacterial cultures will be grown according to standard procedures of the clinical microbiology department. Particularly, the inoculates will be incubated at a temperature of 36 °C for 18 to 24 h, after which the growth of bacteria will be assessed, quantified in CFU/mL, and differentiated. Further culture isolation on a selective Endo-agar medium will be followed by identification and antimicrobial susceptibility testing based on the standards of the European Committee on Antimicrobial Susceptibility Testing (EUCAST).

All isolates will be identified down to a species level through matrix-assisted laser desorption/ionization time-of-flight mass spectrometry (MALDI-TOF MS) on MALDI Biotyper Microflex LT/SH (Bruker, Germany). Susceptibility to antibacterial drugs will be determined by the disk diffusion method. Specifically, Kirby–Bauer agar on plates with dense Mueller–Hinton medium (Gem, Moscow, Russia) and disks with antibiotics (BioRad, Marnes-la-Coquette, France) will be used. The Minimum Inhibitory Concentration (MIC) method of susceptibility testing will be performed using a VITEK 2 Compact 30 analyzer (bioMerieux, Marcy-l’Étoile, France). The EUCAST area of technical uncertainty (ATU) adjustments will be applied for fluoroquinolones in Gram-negative species and clindamycin in *S. aureus*.

The panels of antibiotics used will depend on particular bacterial species, as shown in Table 3.

The results of antimicrobial susceptibility testing will be interpreted in accordance with the criteria of EUCAST version v 15.0 (http://www.eucast.org) or later, if available at the time of testing. An example of a microbiological report form is provided in Appendix A.

#### 2.6.2. Whole-Genome Sequencing

All samples which exhibited detectable bacterial growth and for which the species identification revealed *A. baumannii*, *K. pneumoniae*, *P. aeruginosa*, or *S. aureus* will be subjected to whole-genome sequencing (WGS). If two or more colonies of different bacterial species listed above are isolated from a single sample, then all of them will be subjected to WGS.

The isolates to be sequenced with third-generation sequencing equipment will be selected based on the analysis of short-read sequences and AMR profiles. In particular, a resistance to all classes of antibiotics from the panel and belonging to a global clone of high risk will be among the determining factors. In addition, the most widespread clones of particular species or the ones causing severe infections will also be considered (see Section 2.7 for details).

Genomic DNA of the isolates suitable for cultivation will be isolated with DNeasy Blood and Tissue kit (Qiagen, Hilden, Germany) and used for paired-end library preparation with Nextera™ DNA Sample Prep Kit (Illumina^®^, San Diego, CA, USA) and for MinION library preparation with the Rapid Barcoding Sequencing kit SQK-RBK004 (Oxford Nanopore Technologies, Oxford, UK). Sequencing will be performed on NextSeq (Illumina^®^, San Diego, CA, USA, short reads) and MinION sequencers (Oxford Nanopore Technologies, Oxford, UK, long reads).

An initial read quality check will be made using FastQC v. 0.12 (http://www.bioinformatics.babraham.ac.uk/projects/fastqc, accessed on 11 October 2025). Raw read length and quality filtering will be performed using flexbar 3.5 [38]. The procedures will include Q20 filtering, quality trimming at the end of the reads, removal of library adapter sequences, and removal of A/T stretches and short reads (less than 60 bp).

Base calling of the raw MinION data will be performed with the current version of Guppy Basecalling Software (version 6.4.6), (Oxford Nanopore Technologies, Oxford, UK), and demultiplexing will be performed using Guppy barcoding software version 6.4.6 (Oxford Nanopore Technologies, Oxford, UK). Reads shorter than 1000 bp will be discarded.

Short-read assemblies will be obtained using SPAdes version 4.2.0 [39]. The contigs shorter than 500 bp will be discarded. Hybrid assemblies will be obtained using Unicycler version 0.5.0 [40]. In this case, the contigs shorter than 1000 bp will be discarded. The genomes with an average read coverage lower than 20× will be subjected to recurrent library preparation and sequencing. If the results are not better upon completion of the sequencing, the isolate will be removed from further processing.

#### 2.6.3. Negative Controls and Quality Assurance

The sequencing workflow will incorporate multiple negative controls at each critical stage, such as sample collection, DNA extraction, library preparation, and sequencing. Specifically, uninoculated swabs will be processed identically to patient samples at the same time to detect environmental or cross-contamination during collection procedures. Extraction blanks with no biological material will accompany each batch of samples and will be processed through all molecular steps to identify contamination. Library preparation blanks will be included for each batch of sequencing libraries in a proportion of one negative control per 20 patient samples. Finally, at the sequencing level, negative control libraries will be run on each sequencing flowcell alongside clinical samples, with no reads mapping to target organisms.

Negative control samples that yield any sequence reads from pathogenic bacteria will trigger investigation and sample reprocessing. If multiple negatives fail simultaneously, the entire batch will be re-extracted and re-sequenced from original material. Results from patients whose sequencing runs contained failed negatives will be invalidated.

In order to monitor the barcode leak, i.e., the assignment of reads to incorrect samples, demultiplexing stringency will be set to permit zero or one barcode mismatches, and minimal per-sample sequencing depth will be set to 20. In general, the patient samples should satisfy the following criteria: (1) >80% of reads mapping to target organisms identified in culture; (2) <2% unmapped or ambiguous reads; (3) consistent GC content ± 3% from species reference genome; and (4) no reads matching common environmental contaminants (soil bacteria, fungal sequences, human DNA > 0.1%). The samples failing these thresholds will be retested.

### 2.7. Data Analysis

All the genomes assembled will be processed using a custom software pipeline including a set of scripts for the integration of various available software tools and presenting output in consistent and human-readable formats [21,41]. The scheme of the genomic data analysis pipeline is shown in Figure 2.

The following databases will be used for isolate typing and for revealing possible genes involved in AMR and virulence of the isolates: Resfinder database for AMR gene identification (https://genepi.food.dtu.dk/resfinder, version 4.7.2, identity threshold = 0.8, minimum length threshold = 0.8 [42]), VFDB [43] to search for the virulence factors (http://www.mgc.ac.cn/VFs/main.htm, accessed on 10 October 2025, identity threshold = 0.8, minimum length threshold = 0.8), pubMLST (https://pubmlst.org/, accessed on 10 December 2025) for MLST-typing, and cgMLST (https://www.cgmlst.org/ncs, accessed on 10 October 2025) using MentaList software (https://github.com/WGS-TB/MentaLiST, accessed on 10 October 2025) [44] for cgMLST typing, respectively. The thresholds of cgMLST allele differences to define bacterial clones will be basically as follows: ≤3 for *A. baumannii*, ≤18 for *K. pneumoniae*, ≤37 for *P. aeruginosa*, and ≤15 for *S. aureus* [30]. The cut-off values can be adjusted based on available epidemiological data when necessary.

**Figure 2 mps-08-00151-f002:**
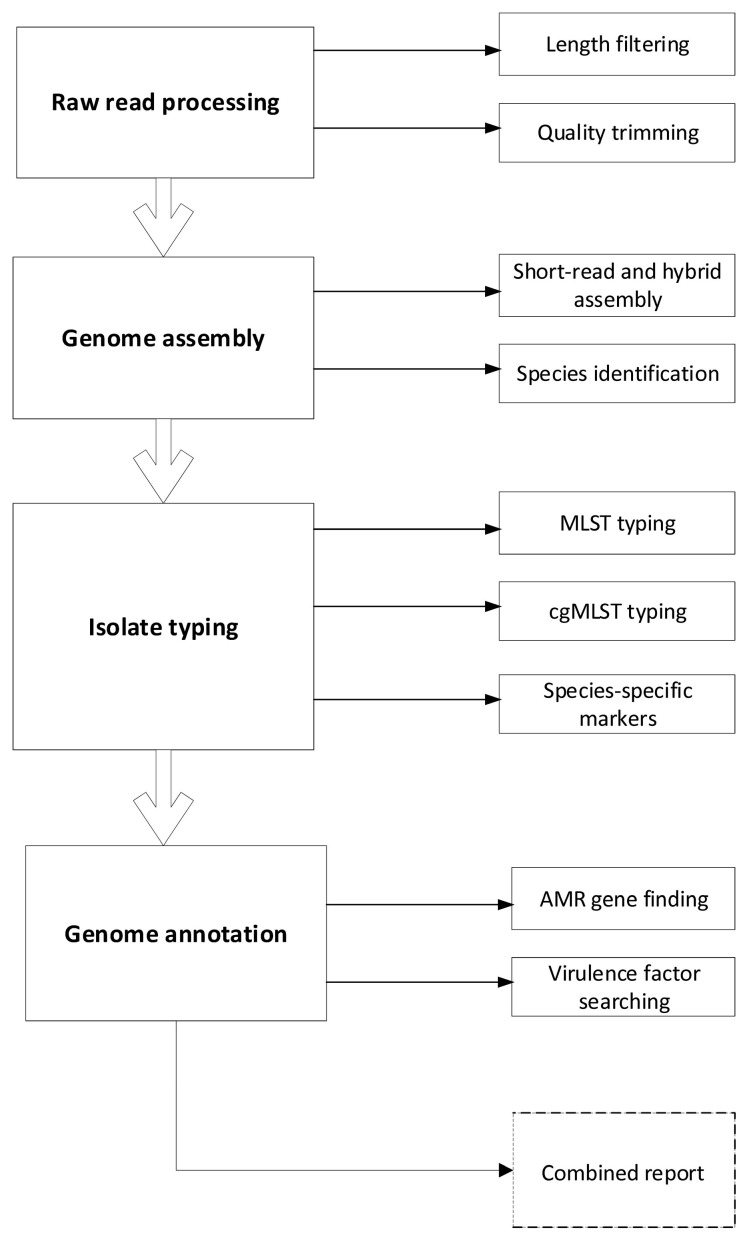
The scheme of genomic data analysis pipeline for the study. Raw sequencing reads undergo quality filtering prior to de novo genome assembly. The resulting assembled genome enables species confirmation, implementation of multiple molecular typing approaches, and comprehensive annotation of antimicrobial resistance determinants and virulence factors. An integrated analytical report consolidating all typing and functional annotation data is generated for each bacterial isolate. MLST—multilocus sequence typing; cgMLST—core genome MLST.

Phylogenetic trees will be built using RAxML (with parameters –f a -m GTRGAMMA –p 12453 -N 1000, i.e., GTRGAMMA model and 1000 bootstrap iterations will be used) [45] and PHYLOViz online (http://online.phyloviz.net, accessed on 10 October 2025). The bootstrap support ≥75% and the number of single nucleotide polymorphisms (SNPs) ≤ 10 will be used to define potential transmission clusters.

The tools and analyses that are specific for particular bacterial species are shown in Table 4.

An important step in the epidemiological surveillance of healthcare-associated infections is pathogen typing, which is greatly facilitated by the availability of genomic data for the isolates. Previously, it was shown that AMR dissemination within the bacterial species belonging to the ESKAPE group is mostly driven by several lineages known as “international clones of high risk” or “global clones” [46,47,48]. Thus, it is essential to determine whether the isolates under study belong to such clones, and this analysis will be performed as a part of the current protocol based on the STs and characteristic genes (e.g., *bla_OXA-51-like_* for *A. baumannii*).

**Table 4 mps-08-00151-t004:** Typing markers and detection software tools to be used in the analysis of bacterial genomes involved in the study.

Species	Additional Isolate Typing Schemes	Programs Used for Typing	Estimated Number of Genomes to Be Sequenced
*A. baumannii*	KL ^1^, OCL ^2^, international clones of high risk (IC)	Kaptive (https://github.com/katholt/Kaptive, accessed on 10 October 2025), custom databases [49]	70
*K. pneumoniae*	KL, OCL, clonal groups (CG)	Kaptive, custom databases [50]	100
*P. aeruginosa*	OCL	PAst (https://github.com/Sandramses/PAst, accessed on 10 October 2025)	60
*S. aureus*	SPA ^3^, clonal complexes (CC)	SPA-typer (https://cge.food.dtu.dk/services/spaTyper/, accessed on 10 October 2025), custom data [48]	20

^1^ KL—capsule synthesis loci, ^2^ OCL—lipooligosaccharide outer core loci, ^3^ SPA—polymorphic VNTR in the 3’ coding region of the *S. aureus*-specific staphylococcal protein A.

In particular, the detection of IC1–IC9 international clones will be performed for *A. baumannii* [46], the assignment to known clonal groups CG20, CG23 (including, among others, ST23-K1 and ST23-K57), CG101, CG147, CG258 (including ST11, ST258, and ST395), CG307, and others [47] will be made for *K. pneumoniae*, clonal complexes CC1, CC5, CC8, CC22, and others will be revealed for *S. aureus* [48], and high-risk clones ST111, ST175, ST233, ST235, ST244, ST654, and others will be detected for *P. aeruginosa* [51,52].

The final typing and annotation report will include sequence type (ST), clonal group/clonal complex/international clone, AMR genes, virulence factors, and specific marker presence for each isolate. An example of such a report is provided in Appendix A.

### 2.8. Statistical Analysis

Statistical analysis will be conducted using R v4.4 (The R Foundation, Austria) and GraphPad Prism V9.1+ (Dotmatics, USA). Depending on data distribution and type, comparisons between colonization profiles in patients with varying pre-existing conditions will be assessed using chi-square tests (for categorical variables), independent *t*-tests, or non-parametric alternatives (e.g., Mann–Whitney U test). To quantify associations, odds ratios will be computed. Additionally, multivariable regression models will evaluate the influence of risk factors, comorbidities, and pathogen characteristics on ICU length of stay and clinical outcomes.

Multicollinearity among severity scores and other correlated variables will be systematically assessed and addressed in multivariable modeling. Variance inflation factors (VIFs) will be computed for all candidate predictors; variables with VIF > 5–10 will be flagged as problematic. Correlation matrices will identify pairs with r > 0.7–0.8, particularly APACHE II and SOFA scores (typically r > 0.80), which cannot be simultaneously included without compromising coefficient stability and interpretability. Domain-based modeling will be employed, constructing separate multivariable models for antibiotic exposure, healthcare exposure, invasive device/support variables, and comorbidities to compare how the inclusion of correlated predictors affects adjusted odds ratios and to assess whether observed changes reflect confounding or collinearity-driven instability.

Where multicollinearity is detected, remediation strategies will be applied according to variable importance. The SOFA score will be retained over APACHE II due to superior specificity for ICU organ dysfunction in colonization contexts. Mechanical ventilation duration will replace binary respiratory support presence to provide a finer risk gradient. Prior antibiotic exposure will be categorized into three levels (none, 3–6 months, >6 months) rather than entered as separate binary variables. Comorbidities will be combined into a Charlson Comorbidity Index or count score to reduce dimensionality while preserving disease burden information. For severity indicators and inflammatory markers (CRP, procalcitonin) with r > 0.50–0.70, only one marker per category will be included; sensitivity analyses will assess robustness to alternative marker selection.

The final multivariable model will be presented with VIF values ≤ 5 for all included variables and diagnostic statistics, confirming acceptable collinearity (condition number < 30, eigenvalues > 0.01). To demonstrate robustness, both a primary model optimized for interpretability with minimal correlated predictors and a sensitivity model with alternative severity score specifications will be reported. Standardized and unstandardized coefficients will be provided; any coefficient shift > 20% with inclusion/exclusion of suspected collinear variables will be explicitly reported and discussed as evidence of model instability. If multicollinearity remains unresolvable, ridge regression or elastic net with cross-validation will be performed as an exploratory alternative to standard logistic regression, trading some interpretability for coefficient stability.

## 3. Expected Results and Outcomes

### 3.1. General Results

The application of the study protocol will allow us to investigate the colonization of ICU patients with AMR pathogens, including the determination of species distribution, phenotypic and genotypic resistance profiles, and the clonal structure of ESKAPE pathogens, as well as to study the changes in these parameters over time. The essential features of this protocol include the WGS of all culturable bacterial isolates regardless of their possible AMR, which allows us to obtain the representative and reliable data on ICU pathogen structure and AMR rates over time.

Specifically, it will be possible to assess the frequency of detection and the colonization pattern by resistant microorganisms for each locus involved. It will also be possible to reveal the most significant risk factors for colonization, including the microbiological profile of the samples taken upon admission from out-of-hospital environments and during a transfer from another department or hospital. In addition, it will be possible to determine the association of different risk factors with colonization patterns upon admission.

The use of the WGS will allow us to study the complete AMR gene repertoire, which is very important for developing prevention measures since some genes might not be expressed at the time of phenotypic resistance profiling, but rather represent a “silent threat”. It will also be possible to reveal the complete clonal structure of a particular pathogen and to compare the isolates collected from different patients to check if they belong to the same strain and/or clone, as well as to compare their AMR gene profiles. Another important output will be the comparison of phenotypic and genomic AMR profiles, which will allow us to reveal possible interpretation errors and, possibly, novel resistance mechanisms.

### 3.2. Stage-Specific Results

During the ICU stay, this protocol will be used to estimate the frequency of detection, the colonization pattern, and the rate of colonization for each locus, which will allow us to determine the starting point of such a colonization. In addition, the influence of risk factors on the colonization rate and pattern for each locus will be assessed. In addition, a relative risk of colonization by AMR microorganisms during the ICU stay will be estimated.

At the stage of completion of hospitalization, a comparison of the rates and patterns of colonization in different loci will be conducted. Moreover, the impact of colonization by AMR microorganisms on the course and outcome of the disease will be determined. Furthermore, the impact of these factors on the outcome of the disease, on the length of stay in the ICU, and on the incidence of nosocomial infection will be estimated.

## 4. Discussion

In recent years, WGS has been extensively used for the epidemiological surveillance of pathogenic bacteria, including outbreak investigations [17,53]. A decision whether a particular isolate belongs to some outbreak or not should be based on some quantitative or qualitative criteria providing sufficient discriminatory power. Although cgMLST or wgMLST-based criteria seem to be a promising option [30], the cutoffs used should probably be flexible to encompass the possible variability of certain species or clones. The studies proposed by us will facilitate developing better cutoff determination strategies and, possibly, will elucidate the mechanisms of bacterial evolution and resistance acquisition, ultimately resulting in outbreaks.

Using WGS data, it will be possible to answer the following questions, which cannot be answered using microbiological studies only. What could be a threshold for distinguishing bacterial strains of the same species (in whole-genome or core genome allele mismatches)? Could these thresholds be reliably defined ad hoc in different situations (e.g., outbreak investigations)? What is the prevalence of multidrug-resistant bacteria in a particular hospital department at particular time point? What is the fraction of bacteria carrying particular resistance determinants? How many genomic mismatches do the isolates belonging to the same species, but obtained from different patients or sources, have? This information will facilitate developing better prevention measures to cope with AMR growth and transfer in clinical conditions.

Currently, the protocol is intended to be applied in one hospital. This hospital could accommodate patients from different regions of the country, but the results obtained from this facility cannot be representative for the whole country or for some region. In addition, the limitation of the study lies in the detection of a certain subset of clinically relevant bacterial species. However, the species intended to be investigated using the protocol were previously shown to be the most widespread in both the study hospital and other multidisciplinary hospitals [35,54]. In general, the protocol is intended to capture the main bacterial colonization pattern in a particular ICU and does not involve the thorough epidemiological surveillance of all the infections revealed in this ICU.

Genomic surveillance of the pathogens causing infections in ICUs was implemented by various researchers worldwide. The recent investigation of *Clostridioides difficile* carriage and transmission in the ICU of a hospital in the USA [55] briefly described a protocol mainly focused on transmission event detection and cannot be extended without significant changes to other species. Another study that involved *A. baumannii* colonization research [56] provided some valuable data, but involved isolates collected more than 8 years ago. A recent study of ICU patient colonization by Gram-negative pathogens [57], surprisingly, did not reveal essential colonization evidence. None of these studies involved the assessment of risk factors for patient colonization. The recent retrospective surveillance of ICU bacterial colonization risk factors from Italy [58] involved such an assessment but did not included genomic data use. Our protocol is intended to bridge the gap between bacterial genomic studies and ICU patient risk factor estimation performed using the epidemiological surveillance not involving WGS.

We are in the process of negotiations with three additional multidisciplinary hospitals and one clinical diagnostics laboratory serving several other hospitals that do not have their own microbiological labs. Additional genomic and phenotypic data obtained from these institutions will facilitate the further refinement of the results of the proposed study.

If the study terminates for some reason (e.g., lack of financing, protocol changes due to pandemic situation), only the obtained data satisfying sufficiency criteria will be processed further. Partial data (e.g., only two genomes of a given species from a patient with a prolonged stay, incomplete assemblies, etc.) will be discarded but will be kept on the institute server for two years in order to be used in the case of study continuation. The results will be published in open access journals to ensure their availability for all interested researchers and clinicians. The genomes obtained will be deposited to Genbank (https://www.ncbi.nlm.nih.gov/genbank/), and possible new sequence type profiles or alleles will be deposited to BIGSdb-Pasteur (https://bigsdb.pasteur.fr/). Bacterial genomic data, the annotation results and associated anonymized clinical data will be kept on the institute server for a period of two years.

Systematic WGS provides an unprecedented capability to identify resistance mechanisms and elucidate transmission pathways of AMR bacteria in intensive care settings. Unlike phenotypic antibiotic susceptibility testing alone, WGS directly identifies specific resistance determinants and reveals the genetic context in which they are carried—whether chromosomal, plasmid-borne, or associated with mobile genetic elements. This genomic characterization enables the detection of inducible or silent resistance mechanisms that may not be apparent from phenotypic testing and allows for discrimination between isolates with phenotypically identical resistance profiles but fundamentally different evolutionary trajectories. Critically, WGS reveals horizontal gene transfer events: multiple isolates from the same patient or different patients carrying identical resistance gene clusters on identical plasmid backbones constitute direct evidence of recent transmission, whereas isolates with the same resistance genes in different genetic contexts suggest independent acquisition or evolutionary divergence. This understanding transforms resistance surveillance from purely descriptive (reporting what an isolate is resistant to) to explanatory (revealing how resistance arose and how it spreads), which is essential for targeted infection control interventions.

The determination of high-risk clonal lineages through genomic analysis carries profound epidemiological and prevention implications. Numerous ESKAPE pathogens are dominated by a limited number of high-risk clones, and WGS enables the rapid assignment of clinical isolates to these lineages and the identification of their characteristic resistance and virulence gene signatures. From an epidemiological perspective, this classification reveals whether a nosocomial infection cluster represents the spread of a known pandemic clone (suggesting ongoing selection pressure and the need for intensive control measures) or the emergence of a novel resistant lineage (suggesting either a new resistance mechanism or unpredicted horizontal gene transfer). From a prevention standpoint, high-risk clone identification allows for risk stratification: patients colonized at ICU admission with a high-risk clone known for enhanced transmissibility, biofilm formation, or virulence can be prioritized for stricter isolation precautions, closer monitoring, or earlier intervention. This precision prevents both resource waste (applying maximal precautions to non-threatening colonization) and missed opportunities (failing to intensify prevention for genuinely dangerous clones).

Longitudinal WGS surveillance during ICU stays captures the dynamic evolution of colonizing clones and the acquisition or loss of resistance genes within individual patients. The detection of progressive resistance acquisition—for example, the accumulation of additional carbapenem-resistance determinants or emergence of variants with enhanced virulence factors—signals ongoing selective pressure from antibiotic therapy and indicates that current infection control strategies are inadequate. Conversely, the persistence of a susceptible-appearing clone despite multiple antibiotic courses may indicate biofilm sequestration or phenotypic tolerance mechanisms not detectable through conventional testing. When multiple patients have genetically close clones in the same ICU timeframe, complete genome sequencing can definitively establish whether this represents a true outbreak (monophyletic transmission cluster), requiring urgent investigation and heightened precautions, or coincidental acquisition of the same globally prevalent clone. This certainty is critical for preventing both outbreak fatigue (over-response to unlinked cases) and missed transmission chains (under-response to genuine nosocomial dissemination).

Collectively, the systematic WGS-based determination of resistance mechanisms and high-risk clones converts ICU colonization surveillance from reactive reporting to proactive epidemiological intelligence, enabling precisely targeted prevention strategies matched to the specific threat profile of detected pathogens.

## Figures and Tables

**Figure 1 mps-08-00151-f001:**
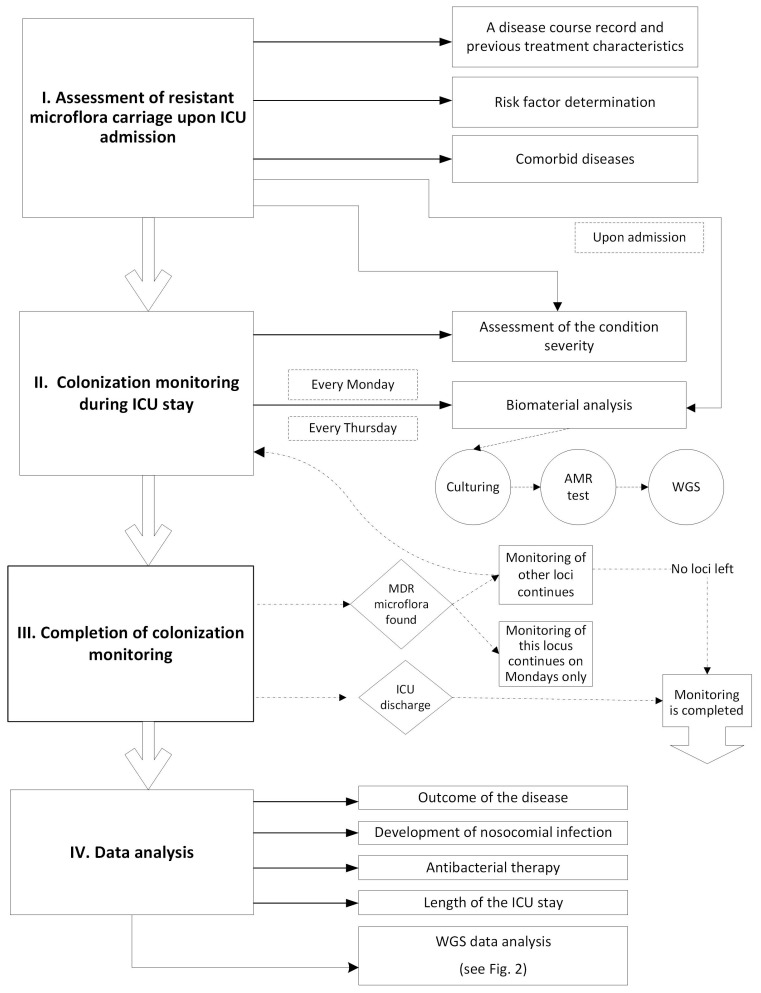
Study design. The study protocol encompasses the following principal phases: (I) baseline microbiological assessment of each patient upon ICU admission to establish the initial colonization profile; (II) longitudinal surveillance of colonization patterns during the ICU stay with predetermined monitoring completion (III) criteria; and (IV) comprehensive patient data analysis following the conclusion of monitoring. The detailed methodology for bacterial genomic data analysis is presented in Figure 2. AMR—antimicrobial resistance; ICU—intensive care unit; MDR—multidrug-resistant (to three or more antibiotic classes); WGS—whole-genome sequencing.

**Table 1 mps-08-00151-t001:** List of inclusion and exclusion criteria.

Inclusion Criteria	Exclusion Criteria
Age over 18	Age under 18
Hospitalization to the ICU	Patients with clinical manifestations of an infection of the upper and/or lower respiratory tract
—	Pregnant women
—	The impossibility of taking a rectal swab and biomaterial from the respiratory tract

**Table 2 mps-08-00151-t002:** Questions/analysis of medical records of all patients upon admission.

Risk Factors	Condition Severity	Previous Treatment	Comorbidities
Hospitalization within previous 3–6 months *	APACHE II score	Rate of viral lung damage (if any)	Oncological diseases
Endoscopy within previous 3 months *	SOFA score	Glucocorticoids taken	Diabetes
Observation in daytime hospital within previous 3 months *	Presence of respiratory support	Antiviral drugs taken	Chronic obstructive pulmonary disease (COPD), bronchial asthma
Taking antibiotics within previous 3 months *	Presence of vasopressor support	Biologic drugs	Chronic heart failure (CHF)
Taking antibiotics within previous 6 months *	—	Bacterial co-infection confirmation (leukocytes, C-reactive protein, procalcitonin, imaging)	Hepatitis, cirrhosis
Living in long-term care facilities *	—	—	Chronic kidney disease (CRF)
Chronic use of glucocorticoids or cytostatics *	—	—	—
Traveling to foreign countries within last month	—	—	—
Presence of indwelling urethral catheter or other invasive devices *	—	—	—
Duration of current hospitalization (days)	—	—	—
Length of ICU stay during current hospitalization (days)	—	—	—
Duration of mechanical ventilation (days) *	—	—	—
Taking antibiotics during current hospitalization *	—	—	—

* specifies primary predictors for resistant microflora colonization in ICU patients.

**Table 3 mps-08-00151-t003:** Antibiotic panels to be used for resistance/susceptibility testing of the bacterial isolates involved in the study.

Species	Antibiotics Panel
*A. baumannii*	amikacin, ciprofloxacin, colistin, gentamicin, imipenem, levofloxacin, meropenem, tobramycin, trimethoprim/sulfamethoxazole
*K. pneumoniae*	amikacin, cefepime, cefotaxime, ceftazidime, ceftazidime/avibactam, ceftriaxone, ciprofloxacin, colistin, fosfomycin, gentamicin, imipenem, meropenem, tobramycin, trimethoprim/sulfamethoxazole
*P. aeruginosa*	amikacin, aztreonam, cefepime, ceftazidime, ceftriaxone, ciprofloxacin, colistin, gentamicin, imipenem, meropenem, netilmicin, tobramycin
*S. aureus*	cefoxitin, clindamycin, gentamicin, erythromycin, linezolid, nitrofurantoin, norfloxacin, oxacillin, rifampicin, teicoplanin, tetracycline, trimethoprim/sulfamethoxazole, vancomycin

## Data Availability

Genomic data for all isolates will be deposited to public databases (Genbank, BIGSdb-Pasteur).

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
