# Peer review of "Study Protocol for Genomic Epidemiology Investigation of Intensive Care Unit Patient Colonization by Antimicrobial-Resistant ESKAPE Pathogens"

_mps, 2025, doi:10.3390/mps8060151_

Round 1
Reviewer 1 Report
Comments and Suggestions for Authors
Shelenkov et al. developed a protocol for hospital application addressing a critical issue, antimicrobial resistance in ESKAPE bacteria. The work is well-structured, thoroughly developed, and includes current and relevant references. The statistical analysis plan is appropriate for this type of study. Furthermore, the study raises a pertinent discussion on the problem of resistance. I recommend publishing after addressing the comments in the minor revisions.
According to the instructions for authors, the study protocols must adhere to the SPIRIT guidelines, but this is not stated in either the abstract or the document.
The abstract is too extended; the maximum is 200 words.
Lines 108 and 232. Do not use the abbreviation Fig. write the whole word Figure.
Line 114. Should "Ethics Approval" go with subheading 2.2?
Line 116 refers to the ethics committee's approval in April 2022, but the tentative recruitment period was scheduled for September 2025 (line 135). This period is long, and there could be changes in treatment protocols and technologies; consider a possible renewal.
The formatting of Table 2 can be improved, as the spaces in the empty columns may cause confusion for the reader. I recommend reorganizing it to improve the formatting.
Lines 160-162. Clarify why monitoring stops when a locus is detected, as this may affect the objective of elucidating the mechanisms of bacterial evolution and the acquisition of resistance, proposed in this work, if sampling stops as soon as an MDR bacterium is found.
In Table 3, in the “Antibiotics panel” for A. baumannii, remove the last comma, and for K. pneumoniae, change the period to a comma after “ceftazidime/avibactam”. Why was colistin used in S. aureus if this bacterium has intrinsic resistance to this antibiotic?
I believe that “Q15 filtering” is a low-quality threshold; this will result in low-quality bases, which can introduce errors in genome assembly and in the identification of resistance genes.
Lines 217 and 221. Do not use “is currently available” or “is available now”, as this is a formal protocol and the available versions must be used.
Author Response
We thank the reviewer for useful comments that has led to significant improvement of our manuscript. Our answers are given below boldfaced.
Shelenkov et al. developed a protocol for hospital application addressing a critical issue, antimicrobial resistance in ESKAPE bacteria. The work is well-structured, thoroughly developed, and includes current and relevant references. The statistical analysis plan is appropriate for this type of study. Furthermore, the study raises a pertinent discussion on the problem of resistance. I recommend publishing after addressing the comments in the minor revisions.
According to the instructions for authors, the study protocols must adhere to the SPIRIT guidelines, but this is not stated in either the abstract or the document.
The instruction for authors states “The study protocols of randomized controlled trials should adhere to the SPIRIT guidelines”. Our protocol does not include randomized controlled studies.
The abstract is too extended; the maximum is 200 words.
We have shortened the abstract accordingly.
Lines 108 and 232. Do not use the abbreviation Fig. write the whole word Figure.
Fixed
Line 114. Should "Ethics Approval" go with subheading 2.2?
We added the subheading 2.2 to Ethics Approval and renumbered the following sections accordingly.
Line 116 refers to the ethics committee's approval in April 2022, but the tentative recruitment period was scheduled for September 2025 (line 135). This period is long, and there could be changes in treatment protocols and technologies; consider a possible renewal.
Initially, the protocol was intended to be applied starting from September 2022, but several non-scientific issues caused by political and financial changes in 2022 have led to a postponement of the study. The treatment protocols and technologies remained the same in general, and minor changes within procedures do not require receiving a new approval according to the effective Order of the Ministry of Health of the Russian Federation dated 1 April 2016 â„–200n “On approval of the rules of good clinical practice”
The formatting of Table 2 can be improved, as the spaces in the empty columns may cause confusion for the reader. I recommend reorganizing it to improve the formatting.
Table 2 was reformatted to increase its readability
Lines 160-162. Clarify why monitoring stops when a locus is detected, as this may affect the objective of elucidating the mechanisms of bacterial evolution and the acquisition of resistance, proposed in this work, if sampling stops as soon as an MDR bacterium is found.
The monitoring was first supposed to reveal only the fact of carbapenem resistance presence. However, later it was changed to continuous monitoring with different intervals for the loci, in which the bacteria have been revealed (once a week vs. twice a week). We updated the manuscript accordingly.
In Table 3, in the “Antibiotics panel” for A. baumannii, remove the last comma, and for K. pneumoniae, change the period to a comma after “ceftazidime/avibactam”. Why was colistin used in S. aureus if this bacterium has intrinsic resistance to this antibiotic?
The panels were harmonized with EUCAST to increase the comparability. The manuscript was updated accordingly
I believe that “Q15 filtering” is a low-quality threshold; this will result in low-quality bases, which can introduce errors in genome assembly and in the identification of resistance genes.
The threshold was changed to Q20, a default one for the filtering programs like FastQC and Trimmomatic. Tightening the threshold further (e.g., Q30) can restrict obtaining the assemblies from the samples of interest, which for some reasons are characterized by relatively low DNA content
Lines 217 and 221. Do not use “is currently available” or “is available now”, as this is a formal protocol and the available versions must be used.
We removed these statements and left the exact versions as suggested
Reviewer 2 Report
Comments and Suggestions for Authors
Reviewer’s Comments and Suggestions for Authors
Journal: Methods and Protocols, MDPI
Manuscript ID: mps-3981557
Type: Study Protocol
Title: Study Protocol for Genomic Epidemiology Investigation of Intensive Care Unit Patient Colonization by Antimicrobial Resistant ESKAPE Pathogens
Authors: Andrey Shelenkov*, Oksana Ni, Irina Morozova, Anna Slavokhotova, Sergey Bruskin, Denis Protsenko, Yulia Mikhaylova and Vasiliy Akimkin
The authors of the Manuscript ID: mps-3981557 constructed a study protocol for genomic epidemiology investigation of intensive care unit patient colonization by antimicrobial resistant ESKAPE pathogens to estimated the risk of AMR infection development and rank its possible severity for the patients admitting ICUs, including analyzing species distribution, assessing phenotypic and genotypic resistance patterns, evaluating the clonal characteristics of ESKAPE pathogens, and tracking temporal variations in these factors.
This manuscript is written clearly and should be interested to readers of the journal. However, essential revisions should be completed as listed below.
Essential revisions
- The Abstract is too redundancy, and should be rewrittenand would be benefited from conciseness.
- In the “Expected Results and Outcomes”section, the authors should divide this section into several subtitles.
- As a study protocol, certain information lacks in the study design or methods, e.g., the samples collected from the patients for the colonization monitoring should be described in the study design. The source of the methods for “Bacteria Cultivation, Identification and Susceptibility Testing” should be cited.
- Lines 274-275: concerning “The essential features of this protocol include the WGS of all culturable bacterial isolates regardless of their possible AMR...”, what did the “all culturable bacterial isolates” mean? Please clarify.
- Figure 1. please rephrase the figure legend.
- Figure 2: concerning “The scheme of genomic data analysis pipeline for the study”, is it different from the routine genomic data analysis protocol?
- Table 3: please fix the “ceftazidime/avibactam. ceftriaxone,”, and check typing errors throughout the manuscript.
- The limitations of the study protocol should be discuss
- Table S1: please format this table, write bacterial species in italics, and fix typing errors.
- References: please carefully check and format the references one by one.
Author Response
We thank the reviewer for useful comments that has led to significant improvement of our manuscript. Our answers are given below boldfaced.
Essential revisions
- The Abstract is too redundancy, and should be rewritten and would be benefited from conciseness.
The Abstract was shortened to 200 words according to the journal requirements.
- In the “Expected Results and Outcomes” section, the authors should divide this section into several subtitles.
Done
- As a study protocol, certain information lacks in the study design or methods, e.g., the samples collected from the patients for the colonization monitoring should be described in the study design. The source of the methods for “Bacteria Cultivation, Identification and Susceptibility Testing” should be cited.
The citations for both local and the corresponding international versions were added
- Lines 274-275: concerning “The essential features of this protocol include the WGS of all culturable bacterial isolates regardless of their possible AMR...”, what did the “all culturable bacterial isolates” mean? Please clarify.
By “culturable isolates” we mean the isolates, for which the standard culturing process allowed to obtain the bacterial colonies to be used in downstream analysis process (resistance testing, DNA isolation etc). If the colonies failed to grow, the sample cannot be processed further and is discarded.
- Figure 1. please rephrase the figure legend.
The extended description and abbreviation explanation were added to the figure captions.
- Figure 2: concerning “The scheme of genomic data analysis pipeline for the study”, is it different from the routine genomic data analysis protocol?
Yes, it is different in performing additional typing procedures like performing cgMLST comparison and more stringent parameters for raw read filtering and phylogenetic analysis (the parameters are described in the section 2.7)
- Table 3: please fix the “ceftazidime/avibactam. ceftriaxone,”, and check typing errors throughout the manuscript.
Fixed
- The limitations of the study protocol should be discuss
Additional statements were added to Discussion section
- Table S1: please format this table, write bacterial species in italics, and fix typing errors.
The table was reformatted to increase the readability. Errors were fixed.
- References: please carefully check and format the references one by one.
The reference, including newly added ones, were checked and formatted according to the journal guidelines.
Reviewer 3 Report
Comments and Suggestions for Authors
Shelenkov et al. propose a study protocol outlining genomic surveillance of antimicrobial-resistant Enterococcus faecium, Staphylococcus aureus, Klebsiella pneumoniae, Acinetobacter baumannii, Pseudomonas aeruginosa, and Enterobacter species in ICU patients. In this protocol, samples collected at admission and twice weekly will undergo culture, species identification, susceptibility testing, and whole-genome sequencing. Genomic data will characterize resistance and virulence determinants, clonal structure, and temporal colonization patterns. Clinical information will be analyzed to identify risk factors for AMR acquisition. Overall, the design is clearly described but lacks sufficient novelty, as it follows already established AMR surveillance approaches, despite offering value through systematic genomic tracking. Therefore, I recommend rejection of the manuscript. I wish the authors all the best with their next submission.
Author Response
Shelenkov et al. propose a study protocol outlining genomic surveillance of antimicrobial-resistant Enterococcus faecium, Staphylococcus aureus, Klebsiella pneumoniae, Acinetobacter baumannii, Pseudomonas aeruginosa, and Enterobacter species in ICU patients. In this protocol, samples collected at admission and twice weekly will undergo culture, species identification, susceptibility testing, and whole-genome sequencing. Genomic data will characterize resistance and virulence determinants, clonal structure, and temporal colonization patterns. Clinical information will be analyzed to identify risk factors for AMR acquisition. Overall, the design is clearly described but lacks sufficient novelty, as it follows already established AMR surveillance approaches, despite offering value through systematic genomic tracking. Therefore, I recommend rejection of the manuscript. I wish the authors all the best with their next submission.
Firstly, the study protocol is intended, by its definition, to clarify the application of the “established approaches” in order to use them in a reproducible and unambiguous way in a daily routine, and not to provide cutting-edge technologies or methods. Second, most existing AMR surveillance approaches do not take into account the genomic data and, especially, the spread of high-risk clones and tracking AMR transmission from this perspective. On the other hand, the studies involving WGS of a large number of isolates usually focus on their genomic characteristics, and do not try to reveal the epidemiologically relevant associations, namely, the influence of bacterial genomic factors on the risk of patient colonization or disease severity. Our protocol provides the insights to bridge this gap.
If the respected reviewer can provide the links to the published protocols describing the same process of colonization monitoring as in our manuscript using the advantages of WGS data and ranging risk factors, we will be happy to add the discussion of their advantages and disadvantages and detailed comparison with our protocol to the manuscript.
We added the description of relevant studies found by us to the Discussion section.
Reviewer 4 Report
Comments and Suggestions for Authors
This study present aa detailed study protocol for genomic epidemiology surveillance of antimicrobial-resistant (AMR) ESKAPR pathogens in ICU patients. This topic is highly relevant, and the proposed workflow is comprehensive, covering sampling, microbiology, whole-genome sequencing and data analysis. This manuscript is designed well and explains an appropriate level of methodological detail. Here are some suggestions to further enhance the work impact even more.
- What is specifically advancement does this protocol offer compared to previous methods?
- How this protocol covers the present gaps in existing ICU AMR surveillance approaches. And add brief comparison existing one with previously established protocol.
- This manuscript stated 100 patients, but no statistical rationale is provided, and it is not clear whether this number of patients are sufficient to detect meaningful difference in colonization rate or risk factors.
- How patient data will be anonymized and the duration of storing genomic and clinical data please clarify.
- Explain any genomic data could pose privacy risk and how these will be mitigated.
- Figures 1 and 2 are useful but more descriptive caption and clarification of all abbreviations needed.
- Please check the manuscript to maintain consistency spacing , especially around parentheses and hyphens, and fix some words are splits incorrectly e.g. pa ern", "se ings.
- It is better to introduce all abbreviations at first use MDR<CRF<ICU and please maintain consistency use of AMR vs MDR through the manuscript.
- Some references are missing pages or complete details. So please revise them to follow journal format.
Author Response
We thank the reviewer for useful comments that has led to significant improvement of our manuscript. Our answers are given below boldfaced.
This study present aa detailed study protocol for genomic epidemiology surveillance of antimicrobial-resistant (AMR) ESKAPR pathogens in ICU patients. This topic is highly relevant, and the proposed workflow is comprehensive, covering sampling, microbiology, whole-genome sequencing and data analysis. This manuscript is designed well and explains an appropriate level of methodological detail. Here are some suggestions to further enhance the work impact even more.
- What is specifically advancement does this protocol offer compared to previous methods?
- How this protocol covers the present gaps in existing ICU AMR surveillance approaches. And add brief comparison existing one with previously established protocol.
1&2 - Our protocol is intended to bridge the gap between the bacterial genomic studies and the ICU patient risk factor estimation performed using the epidemiological surveillance not involving WGS. We added the description to the Discussion section. In Russia, currently there is not existing medical protocol for studying the ICU patient colonization by bacterial pathogens involving the use of WGS data. Although various investigations of ICU patients using WGS were performed worldwide, we have not yet found the detailed protocol describing the ranging of risk factors with application of WGS data for this purpose.
3. This manuscript stated 100 patients, but no statistical rationale is provided, and it is not clear whether this number of patients are sufficient to detect meaningful difference in colonization rate or risk factors.
The results of epidemiological studies of infections in an ICU of the same hospital (doi: 10.36488/cmac.2024.2.124140) provided some baseline data for the current protocol. However, since some variation could occur in later studies and this protocol can be later applied in other hospitals, we increased the number of patients to 250 to ensure the obtainment of sufficient amounts of data. However, this value provides the lower boundary, and all patients satisfying the criteria can be included in the study during the specified period.
4. How patient data will be anonymized and the duration of storing genomic and clinical data please clarify.
The unique alpha-numeric code will be assigned to each participating patient upon first sample obtainment. Since that, all microbiological reports will contain this code, gender and age. The researchers involved in data processing will not have access to the patients’ personal data.
Bacterial genomic data, the annotation results and associated anonymized clinical data will be kept on the institute server for the period of two years (see Discussion section).
5. Explain any genomic data could pose privacy risk and how these will be mitigated.
Genomic data used in the current study could not pose privacy risk since only the bacterial genomic data is collected and no human genome data will be used
6. Figures 1 and 2 are useful but more descriptive caption and clarification of all abbreviations needed.
The extended description and abbreviation explanation were added to the figure captions.
7. Please check the manuscript to maintain consistency spacing , especially around parentheses and hyphens, and fix some words are splits incorrectly e.g. pa ern", "se ings.
The manuscript was checked to remove formatting errors
8. It is better to introduce all abbreviations at first use MDR<CRF<ICU and please maintain consistency use of AMR vs MDR through the manuscript.
The abbreviations were defined at first use, and the use of AMR (antimicrobial resistance) vs. MDR (multidrug resistance, i.e., resistance to three or more antibiotic classes) was corrected.
9. Some references are missing pages or complete details. So please revise them to follow journal format.
The reference, including newly added ones, we checked and formatted according to the journal guidelines.
Reviewer 5 Report
Comments and Suggestions for Authors
The manuscript presents a detailed and well-structured study protocol aimed at applying whole-genome sequencing–based genomic epidemiology to monitor ICU colonization by antimicrobial-resistant ESKAPE pathogens. This is a timely and highly relevant topic, and the authors provide an extensive methodological description.
Nevertheless, several aspects could be refined to further improve clarity, reproducibility, and scientific value.
Major Comments
- Study design clarity
The overall design (admission → biweekly sampling → discharge evaluation) is conceptually strong. However, several elements of the design would benefit from additional clarification:
-
Stopping sampling after detection of MDR from a given locus may limit longitudinal interpretation. Providing justification (e.g., resource allocation, redundancy based on previous surveillance data) would strengthen this section. The current rule may bias temporal comparisons if MDR is detected early.
-
The 3-month data collection window might be insufficient to capture seasonal variation or atypical colonization patterns in ICUs, especially for organisms with low prevalence. The authors mention possible extension, but the rationale for the initial 3-month period should be explained.
-
- Representativeness and sample size
The protocol specifies 100 patients. It would be useful to include:
-
An a priori calculation or justification for why this number is sufficient for statistical analyses proposed in Section 2.7.
-
Expected prevalence ranges of the ESKAPE species in the target ICU, as this would influence the number of genomes sequenced per species (Table 4).
-
- Sequencing strategy and selection criteria
The WGS pipeline is well described. However, the selection criteria for long-read sequencing require further specification:
-
How will the authors define “global high-risk clones” for each species (e.g., for K. pneumoniae: ST11, ST258, ST307, ST23 hypervirulent lineages)?
-
If selection is based on phenotypic MDR profile, the protocol may underrepresent ESBL-only isolates or susceptible strains, which may be relevant for local epidemiology.
-
- Data analysis: Need for reproducibility details
The genomic analysis workflow is sufficiently detailed but the following points require clarification:
-
Will raw sequencing reads be quality-checked with tools such as FastQC before trimming?
-
For phylogenetic analysis, it would be important to specify:
-
which RAxML model will be used (GTRGAMMA or alternative),
-
whether bootstrapping will be performed,
-
thresholds for defining potential transmission clusters.
-
-
- Risk factor assessment
Table 2 lists extensive variables, but:
-
It is unclear which variables will be treated as primary vs secondary predictors.
-
Handling of missing data (common in ICU retrospective variables) should be outlined.
-
For multivariable models, potential multicollinearity (e.g., SOFA and APACHE II scores together) should be considered.
-
Minor Comments
-
Figure 1 and Figure 2 are clear, but captions should be expanded to fully describe each step so the figures are interpretable without referencing the main text.
-
In Section 2.5.1, specify whether EUCAST area of technical uncertainty (ATU) adjustments will be applied.
-
Consider adding a short subsection on contamination control in sequencing workflows (negative controls, barcode leak monitoring), as this is essential in ICU genomic epidemiology.
-
The manuscript mentions dissemination to additional hospitals; this could be moved to Discussion to avoid mixing future plans with protocol description.
-
Consider harmonizing antibiotic panels with EUCAST recommended panels for better international comparability.
This is a well-constructed and timely protocol with high potential impact for genomic surveillance in ICU settings. The methodological rigor is strong, and the pipeline is consistent with current best practices. The suggested clarifications will improve reproducibility and strengthen the protocol’s generalizability.
Author Response
The manuscript presents a detailed and well-structured study protocol aimed at applying whole-genome sequencing–based genomic epidemiology to monitor ICU colonization by antimicrobial-resistant ESKAPE pathogens. This is a timely and highly relevant topic, and the authors provide an extensive methodological description.
Nevertheless, several aspects could be refined to further improve clarity, reproducibility, and scientific value.
We thank the reviewer for useful comments that has led to significant improvement of our manuscript. Our answers are given below boldfaced.
Major Comments
- Study design clarity
The overall design (admission → biweekly sampling → discharge evaluation) is conceptually strong. However, several elements of the design would benefit from additional clarification:
- Stopping sampling after detection of MDR from a given locus may limit longitudinal interpretation. Providing justification (e.g., resource allocation, redundancy based on previous surveillance data) would strengthen this section. The current rule may bias temporal comparisons if MDR is detected early.
The monitoring was first supposed to reveal only the fact of carbapenem resistance presence. However, later it was changed to continuous monitoring with different intervals for the loci, in which the bacteria have been revealed (once a week vs. twice a week). We updated the manuscript accordingly.
- The 3-month data collection window might be insufficient to capture seasonal variation or atypical colonization patterns in ICUs, especially for organisms with low prevalence. The authors mention possible extension, but the rationale for the initial 3-month period should be explained.
The 3-month period was chosen since such a duration was likely to allow recruiting sufficient number of participants according to the previous ICU occupation and stay duration data, which was mentioned in the manuscript. We added more explicit statement to reflect this fact. In our previous short-term studies, we have not revealed a significant seasonal variation in bacterial colonization patterns in ICU, although such variation could in fact take place. At the same time, the data collection period can be extended if some unusual variations occur because of this or some other factors
- Representativeness and sample size
The protocol specifies 100 patients. It would be useful to include:
- An a priori calculation or justification for why this number is sufficient for statistical analyses proposed in Section 2.7.
The results of epidemiological studies of infections in an ICU of the same hospital (doi: 10.36488/cmac.2024.2.124140) provided some baseline data for the current protocol. However, since some variation could occur in later studies and this protocol can be later applied in other hospitals, we increased the number of patients to 250 to ensure the obtainment of sufficient amounts of data. However, this value provides the lower boundary, and all patients satisfying the criteria can be included in the study during the specified period.
- Expected prevalence ranges of the ESKAPE species in the target ICU, as this would influence the number of genomes sequenced per species (Table 4).
The prevalence ranges for the pathogens were previously calculated based on the patient infection data (doi: 10.36488/cmac.2024.2.124140, unpublished data – in this study, the samples were taken from blood and revealed infection loci), but it is very complicated to reliably estimate the colonization by this pathogens since the colonized patients might not exhibit any infection symptoms. We used the reference data from the Russian Ministry of Health and Science report for the year 2024, which included some averaged data regarding the pathogen prevalence ranges in ICUs.
- Sequencing strategy and selection criteria
The WGS pipeline is well described. However, the selection criteria for long-read sequencing require further specification:
- How will the authors define “global high-risk clones” for each species (e.g., for K. pneumoniae: ST11, ST258, ST307, ST23 hypervirulent lineages)?
The definition of global high-risk clones for A. baumannii, K. pneumoniae and S. aureus is made using previously published data. The description of specific clones and references were added to the manuscript (section 2.7)
- If selection is based on phenotypic MDR profile, the protocol may underrepresent ESBL-only isolates or susceptible strains, which may be relevant for local epidemiology.
The selection for long-read sequencing will be based on several criteria, including phenotypic profile, belonging to high-risk clones and the presence of genomic resistance determinants. The goal of the protocol is to study the colonization patterns by resistant bacteria, and the epidemiological surveillance of a full bacterial specter (including fully susceptible isolates) lies beyond the scope of the protocol. The epidemiological study of infection prevalence was performed previously in the same hospital (doi: 10.36488/cmac.2024.2.124140).
- Data analysis: Need for reproducibility details
The genomic analysis workflow is sufficiently detailed but the following points require clarification:
- Will raw sequencing reads be quality-checked with tools such as FastQC before trimming?
Yes, the FastQC will be used. We added this point to the description
- For phylogenetic analysis, it would be important to specify:
- which RAxML model will be used (GTRGAMMA or alternative),
- whether bootstrapping will be performed,
- thresholds for defining potential transmission clusters.
The details were added to the manuscript, section 2.7. Briefly, GTRGAMMA model is used, bootstrapping is performed with 1000 iterations, and the threshold of 75% bootstrap support and <=10 SNPs will be used.
- Risk factor assessment
Table 2 lists extensive variables, but:
- It is unclear which variables will be treated as primary vs secondary predictors.
The description was added to the manuscript
- Handling of missing data (common in ICU retrospective variables) should be outlined.
The description of handling missing data was added to Section 2.5.1 just after Table 2
- For multivariable models, potential multicollinearity (e.g., SOFA and APACHE II scores together) should be considered.
The description was added to statistical analysis section (2.8)
Minor Comments
- Figure 1 and Figure 2 are clear, but captions should be expanded to fully describe each step so the figures are interpretable without referencing the main text.
The extended description and abbreviation explanation were added to the figure captions.
- In Section 2.5.1, specify whether EUCAST area of technical uncertainty (ATU) adjustments will be applied.
The statement was added to section 2.6.1 (updated number due to section addition) just before the Table 3.
- Consider adding a short subsection on contamination control in sequencing workflows (negative controls, barcode leak monitoring), as this is essential in ICU genomic epidemiology.
The section 2.6.3 was added.
- The manuscript mentions dissemination to additional hospitals; this could be moved to Discussion to avoid mixing future plans with protocol description.
Revised as suggested
- Consider harmonizing antibiotic panels with EUCAST recommended panels for better international comparability.
The panels were harmonized with EUCAST to increase the comparability.
This is a well-constructed and timely protocol with high potential impact for genomic surveillance in ICU settings. The methodological rigor is strong, and the pipeline is consistent with current best practices. The suggested clarifications will improve reproducibility and strengthen the protocol’s generalizability.
Round 2
Reviewer 2 Report
Comments and Suggestions for Authors
Reviewer’s Comments and Suggestions for Authors
(Second Round)
Journal: Methods and Protocols, MDPI
Manuscript ID: mps-3981557-R1
Type: Study Protocol
Title: Study Protocol for Genomic Epidemiology Investigation of Intensive Care Unit Patient Colonization by Antimicrobial Resistant ESKAPE Pathogens
Authors: Andrey Shelenkov*, Oksana Ni, Irina Morozova, Anna Slavokhotova, Sergey Bruskin, Denis Protsenko, Yulia Mikhaylova and Vasiliy Akimkin
My concerns raised in the previous reviewing of the Study Protocol Manuscript ID:mps-3981557 have been adequately addressed. The manuscript has been substantially improved. In my opinion, the work could be published.
Author Response
Thank you for reviewing the manuscript.
Reviewer 3 Report
Comments and Suggestions for Authors
My decision on the manuscript, as communicated in the first round of review and in my email to the editor, remains unchanged. In its current form, the work lacks novelty and the supporting results needed to assess the protocol's utility, and thus its contribution to the field is very limited.
Author Response
The manuscript now includes a dedicated description of the advantages of applying WGS to studies of bacterial colonization in the ICU. However, because neither the specific limitations of the proposed protocol nor the strengths and representative examples of alternative approaches were currently described by the respected reviewer, the scope for a balanced and rigorous scientific discussion is markedly constrained.
Our description is provided below:
Systematic WGS provides unprecedented capability to identify resistance mechanisms and elucidate transmission pathways of antimicrobial-resistant (AMR) bacteria in intensive care settings. Unlike phenotypic antibiotic susceptibility testing alone, WGS directly identifies specific resistance determinants and reveals the genetic context in which they are carried—whether chromosomal, plasmid-borne, or associated with mobile genetic elements. This genomic characterization enables detection of inducible or silent resistance mechanisms that may not be apparent from phenotypic testing and allows discrimination between isolates with phenotypically identical resistance profiles but fundamentally different evolutionary trajectories. Critically, WGS reveals horizontal gene transfer events: multiple isolates from the same patient or different patients carrying identical resistance gene clusters on identical plasmid backbones constitute direct evidence of recent transmission, whereas isolates with the same resistance genes in different genetic contexts suggest independent acquisition or evolutionary divergence. This understanding transforms resistance surveillance from purely descriptive (reporting what an isolate is resistant to) to explanatory (revealing how resistance arose and how it spreads), which is essential for targeted infection control interventions.
The determination of high-risk clonal lineages through genomic analysis carries profound epidemiological and prevention implications. Numerous ESKAPE pathogens are dominated by a limited number of high-risk clones, and WGS enables rapid assignment of clinical isolates to these lineages and identification of their characteristic resistance and virulence gene signatures. From an epidemiological perspective, this classification reveals whether a nosocomial infection cluster represents spread of a known pandemic clone (suggesting ongoing selection pressure and need for intensive control measures) or emergence of a novel resistant lineage (suggesting either a new resistance mechanism or unpredicted horizontal gene transfer). From a prevention standpoint, high-risk clone identification allows risk stratification: patients colonized at ICU admission with a high-risk clone known for enhanced transmissibility, biofilm formation, or virulence can be prioritized for stricter isolation precautions, closer monitoring, or earlier intervention. This precision prevents both resource waste (applying maximal precautions to non-threatening colonization) and missed opportunities (failing to intensify prevention for genuinely dangerous clones).
Longitudinal WGS surveillance during ICU stay captures the dynamic evolution of colonizing clones and the acquisition or loss of resistance genes within individual patients. Detection of progressive resistance acquisition—for example, accumulation of additional carbapenem-resistance determinants or emergence of variants with enhanced virulence factors—signals ongoing selective pressure from antibiotic therapy and indicates that current infection control strategies are inadequate. Conversely, persistence of a susceptible-appearing clone despite multiple antibiotic courses may indicate biofilm sequestration or phenotypic tolerance mechanisms not detectable by conventional testing. When multiple patients genetically close clones in the same ICU timeframe, complete genome sequencing can definitively establish whether this represents a true outbreak (monophyletic transmission cluster) requiring urgent investigation and heightened precautions, or coincidental acquisition of the same globally-prevalent clone. This certainty is critical for preventing both outbreak fatigue (over-response to unlinked cases) and missed transmission chains (under-response to genuine nosocomial dissemination).
Collectively, systematic WGS-based determination of resistance mechanisms and high-risk clones converts ICU colonization surveillance from reactive reporting to proactive epidemiological intelligence, enabling precisely targeted prevention strategies matched to the specific threat profile of detected pathogens.